# Using data derived from cellular phone locations to estimate visitation to natural areas: An application to water recreation in New England, USA

**Nathaniel H. Merrill**[1]*, **Sarina F. Atkinson**[2], **Kate K. Mulvaney**[1], **Marisa J. Mazzotta**[1], **Justin Bousquin**[3]

**1** Atlantic Coastal Environmental Sciences Division, U.S. Environmental Protection Agency, Office of Research and Development, Center for Environmental Measurement and Modeling, Narragansett, Rhode Island, United States of America, **2** Cooperative Institute for Marine & Atmospheric Studies, Rosenstiel School of Marine & Atmospheric Science, University of Miami, Miami, Florida, United States of America, **3** Gulf Ecosystem Measurement and Modeling Division, U.S. Environmental Protection Agency, Office of Research and Development, Center for Environmental Measurement and Modeling, Gulf Breeze, Florida, United States of America

* merrill.nathaniel@epa.gov

**Data Availability Statement:** All relevant data are available at https://github.com/USEPA/Recreation_Benefits.git.

## Abstract

We introduce and validate the use of commercially available human mobility datasets based on cell phone locations to estimate visitation to natural areas. By combining this data with on-the-ground observations of visitation to water recreation areas in New England, we fit a model to estimate daily visitation for four months to more than 500 sites. The results show the potential for this new big data source of human mobility to overcome limitations in traditional methods of estimating visitation and to provide consistent information at policy-relevant scales. However, the data providers' opaque and rapidly developing methods for processing locational information required a calibration and validation against data collected by traditional means to confidently reproduce the desired estimates of visitation. We found that with this calibration, the high-resolution information in both space and time provided by cell phone location-derived data creates opportunities for developing next-generation models of human interactions with the natural environment.

## Introduction

People visit natural areas to view the scenery and wildlife or to engage in any number of recreational activities they enjoy. These areas are important to society, as shown through the sheer number of people who visit parks, beaches, walking trails, and other natural spaces and their significant contribution to the economy [1]. However, it is difficult to quantify the use of natural areas across the many, diverse locations and the timing of those visits. Observational counts are time consuming and expensive to conduct and traditional survey approaches have their own sampling complications for estimating visitation. Therefore, it is often unknown how

**Funding:** The author(s) received no specific
funding for this work.

**Competing interests:** The authors have declared
that no competing interests exist.

many and what types of people visit natural areas—critical information for managers or
researchers to apply in natural resource damage assessments, park and urban planning, eco-
nomic valuation studies, tourism studies, as well as to inform many other management deci-
sions [2–3].

This paper demonstrates the use of commercially available anonymized and aggregated
data on cellular device locations to estimate visitation to natural areas. For brevity, we refer to
these cellular device location-based datasets generally as "cell data" in this paper. We investi-
gated how cell data performs in providing the types of visitation information needed in policy
applications, information on the temporal and spatial distribution of visits to natural areas.
Specifically, we compared the cell data with on-the-ground daily visitation counts that we col-
lected, along with other federal, state and town records for water recreation areas. We found
that the cell data contained useful information for estimating visitor use, but it required a cor-
rection (calibration) to match the scale of the observations and to be confident with its wider
application. We then built and applied a statistical model to estimate daily visitation with cell
data to more than 500 water recreation areas in the northeastern United States for the primary
four months of recreational use (June-September) of 2017.

Our application demonstrates the potential for this emerging source of big data to provide
comprehensive visitation information across many places and time windows, a feat that would
be impractical with traditional methods. The high-resolution information in space and time
provided by cell data expands opportunities for developing next-generation models of human
interactions with the natural environment. In addition, cell data provides the ability to know
not just how many people are visiting specific locations, but also where visitors are coming
from within aggregated geographies. This information allows for basic calculations of distance
traveled to a location and a deeper understanding of the community composition of visitors.
While these new data sources may help overcome many logistical barriers to obtaining behav-
ioral information at scale, our work highlights the ongoing need for traditional methods of col-
lection for calibration and validation for these new data sources to be useful in common
applications.

## Background

Existing visitation information for natural areas is limited and currently comes from many dif-
ferent, often inconsistent, sources. There are visitation estimates derived from entrance fees,
parking fees, lifeguard counts, car or people electronic counters, aerial surveys, remote sensing,
or tailored observational sampling plans [4–14]. Each data source comes with its own nuances
in terms of sampling issues and geographical and temporal coverage. Most ongoing visitation
collection efforts only capture paying customers during the hours of fee collections, thereby
providing only a subset of daily use. There are also detailed collections that coincide with a spe-
cific project or event, such as an oil spill, but the knowledge is often not generalizable past a
particular region or time-period. Often, the need to obtain visitation information arises after
an event has occurred, making the before and after comparison difficult [3]. These event-
based data collections are also resource intensive [2].

Efforts to overcome these barriers using other types of digital records include estimates
derived from photo-sharing or other types of social media posts [15–21]. These techniques
have been useful in estimating visitation to large parks, attractions, and natural areas around
the world over long periods of time by providing monthly, seasonal, or yearly visitation esti-
mates. However, these data sources represent only the small fraction of the public that opt to
use those specific social media outlets, and they lack adequate temporal and spatial resolution.
These factors limit the ability of social-media based methods to inform broader policies or

localized environmental management. Use of cellular devices is much more common and the resolution of the information provided is much finer in both time and space.

Cell data come from the digital traces of people's cell phone use and location. In the past, this was based on the location of the tower a device was communicating with or a triangulation from the device to various cellphone towers. This data provided information based on many people, but with locational and temporal accuracy issues related to connectivity to the cellphone tower network and peoples' active use of the phone [22]. With the addition of GPS instrumentation and the growing ubiquity of smartphone-style cellular telephones, these digital traces have become more accurate, more frequent and represent an increasing proportion of the population [23–28]. The device-level, raw data are collected by cellular telephone providers, GPS enabled devices and increasingly by smartphone applications. Several third-party providers on the commercial market combine this information and sell processed data in a variety of formats [28].

Cell data in various forms have been used most widely in the transportation and urban planning fields to understand use of public infrastructure and commuting [23–26] and to assist in land use classifications [27]. This data has also been used to understand economic trends [29], restaurant choice [30] and in epidemiology and population research in developing countries [31–34]. Despite its promise, there are limited environmental applications of the data to date, with notable applications concerning natural disasters [32] and air quality [35–36].

There are a few recent applications of cell data to understand behavior in and around natural areas [37–39]. Kubo et al. [37] used cell data to calculate the economic value of coastal tourism across Japan, but provided no ground truth to the visitation information. For an island park in Korea [38], Kim et al. applied cell data to analyze tradeoffs between visitation and biodiversity and showed decent correlations between the cell data and monthly estimates of visitation to several specific locations on the island. A study of parks in California, USA, is the closest to the work presented in this paper [39]. The study used a similarly processed cell data product from a third-party vendor to estimate daily park visitation. They calibrated the cell data with just one set of data, vehicle counts on a major nearby road, finding a unit-value correction factor. They then validated their estimates against a single park's gate traffic and parking information. They found good agreement with their corrected cell data model and daily vehicle counts. From there, they used park-specific vehicle-to-people ratios to extrapolate to the number of visitors to the other twenty-one parks of interest. Our study differs by incorporating multiple visitation records representing counts of people to a wider set of locations: eighteen different water recreation areas. We find similar potential for this data source to provide useful, policy-relevant visitor use information at daily and site-level scales for water recreation areas.

## Data description

We identified and purchased a dataset of visitation derived from cell phone locations for a set of geographies and time to understand the extent, temporal distribution, and value of water recreation for Cape Cod, Barnstable County, Massachusetts and New England, USA in general. The set of water recreation areas comprises a comprehensive list of all the public beaches and public access points to water (fresh and saltwater beaches, public access points, parks, ways to water, and boat ramps) for Barnstable County, Massachusetts, compiled from federal, state, county, and town GIS information (n = 464), and an additional set of beaches across greater New England (n = 113). The data we used consists of estimates of visitation to these water recreation areas over the summer months of 2017 (June through September).

We purchased data products processed by a third-party provider, Airsage, Inc. This provider creates population-level estimates of human mobility derived from a panel of over 120 million devices using location information from smartphone applications (see S1 File). The data provider processes this device-specific locational information. Before we receive it, the data is anonymized and aggregated to contain no personally identifiable information. We do not obtain any device-level information, nor raw device GPS locations, but instead, we obtain aggregated summaries of visitation by recreation site and estimates of the visitors' origin census block-group geographies. The data provider translates their sample to population-level estimates using weights based on the share of the population their sample represents by census-tract geographies. The cellular device sample we purchased data from includes about 30% of the U.S. population but varies by tract and month.

To obtain the cell data for the sample geographies of interest, we spatially buffered (added area) around the water-access sites which were designated as line or point features in the original spatial databases. In consultation with the data provider and after attempting a range of spatial buffers, a 100-meter buffer was chosen to balance specificity in capturing water recreation visits (i.e., not capturing ancillary points of interest in geographies, like restaurants or stores, for example) with the accuracy of the locational information. We sent the defined water recreation areas to the data provider as a set of geographic extents, or polygons (see Fig 1 for examples of area definitions), and they returned the aggregated and anonymized processed data in tabular form. We include a sample of the dataset below (Table 1) and include the entirety of this dataset available with the code package associated with this work at https://github.com/USEPA/Recreation_Benefits.git.

The locational accuracy of the device locations underpinning the data range depending on the source device and the smartphone application. The accuracy of reported locations from applications varies with ranges of 1–10 meters (GPS), 20–200 meters (Wi-Fi), and 100–2000 meters (cell tower-based) based on the method(s) each application uses to locate each device. We were not able to obtain an average locational accuracy for devices seen in our geographies in our specific dataset since the smartphone applications do not report the exact location methods to the data provider and we do not receive device-specific locational information. Given the potential range in location accuracy, visits attributed to a water recreation area could have actually been to a nearby attraction, or vice-versa. We chose a relatively small buffer around the recreation areas to be conservative in defining the area attributed to use of the site and to minimize any mis-located visits. Given this and other limitations, we relied on the calibration and validation to on-the-ground visitation counts to assess the usefulness and accuracy of the cell data for our application and the choice of spatial definitions and buffers around sites.

In total, the cell dataset includes visitation estimates for 51,511 days across 577 sites. A complete set would be 70,394 days (577 sites x 122 days), but some of the days for some sites are missing due to low visitation and detection limits. A visit was defined as a device location history implying a stay estimated to be longer than five minutes at a geography that was not the home or work location of the device based on the behavior of that device over the month. Home was defined as the census block group where the device is most often seen over the month between 9PM-6AM (see S1 File).

In addition to estimates of visits, the cell data for each area also includes the home location of those visitors, either at the census block-group level or categorized as international if the origin was outside of the United States. For example, the data may show 100 people visited beach x during a time period, with 30 of those people come from census block y, 20 from census block z, and so on. The monthly visitor origin-destination data contains 642,915 data points representing monthly trip totals (577 sites x origin census block groups, which vary by site and

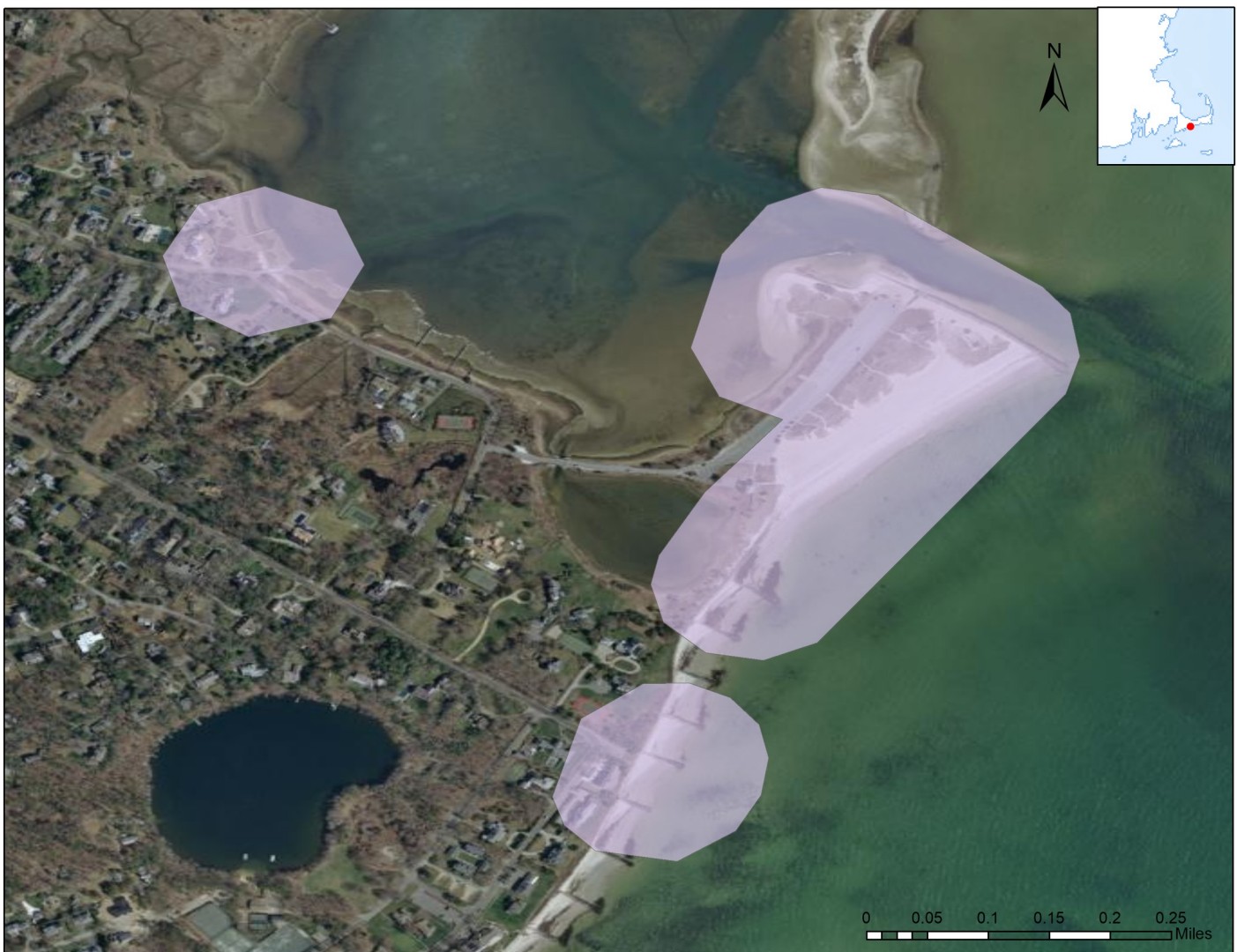

**Fig 1. Example definition of water recreation areas.** Dowses Beach, Barnstable, Massachusetts, USA and nearby water access areas. Point and line features representing water recreation access were buffered by 100 meters to capture use at the sites. These geographic areas correspond to the sample of visitation estimates from the cell data.

month). This total is not inclusive of the data representing zero trips to destinations originating from block groups implied by the full origin-destination matrix. Because of the geographic and temporal scope, collecting this same information with traditional methods would be prohibitively expensive, time consuming, and inconsistent.

## Methods

To investigate the cell data's ability to reproduce daily visitation counts, we compared the cell data to a series of observational counts collected from three different, commonly recorded sources for beach and park visitations. We then calibrated a model to translate the cell data into consistent estimates of visitation across many sites in the region of interest and across many days. We purposely designed the calibration to cover a wide range of access location sizes and visitation totals to test the transferability of the cell data and models built from it. No

**Table 1. Sample of dataset derived from cell phone locations.**

| POI | DATE | HH00 | HH01 | HH02 | HH03 | HH04 | HH05 | . . . | HH20 | HH21 | HH22 | HH23 | DEVICE_TOTAL |
|---|---|---|---|---|---|---|---|---|---|---|---|---|---|
| 1 | 20170601 | 251 | 96 | 47 | 0 | 171 | 488 | . . . | 668 | 848 | 812 | 222 | 21895 |
| 1 | 20170602 | 245 | 202 | 133 | 148 | 112 | 646 | . . . | 594 | 604 | 1157 | 468 | 21526 |
| 1 | 20170603 | 299 | 148 | 196 | 243 | 340 | 135 | . . . | 922 | 1283 | 1100 | 759 | 19449 |
| 1 | 20170604 | 658 | 332 | 117 | 372 | 414 | 395 | . . . | 1071 | 186 | 404 | 286 | 19415 |
| 2 | 20170601 | 0 | 0 | 0 | 0 | 0 | 0 | . . . | 86 | 0 | 0 | 0 | 1103 |
| 2 | 20170602 | 114 | 0 | 0 | 0 | 0 | 50 | . . . | 138 | 312 | 0 | 48 | 2746 |
| 2 | 20170603 | 0 | 0 | 0 | 0 | 0 | 0 | . . . | 238 | 226 | 122 | 66 | 2247 |
| 2 | 20170604 | 66 | 66 | 66 | 66 | 0 | 0 | . . . | 156 | 33 | 0 | 54 | 2046 |

POI indexes the water access locations and each column represents estimated visitation in hourly windows, with HH00 being 12AM-1AM. DEVICE_TOTAL refers to the estimate of unique devices seen in any of the 24 hours at that water access location.

visitation dataset alone was perfect for use in calibration across a wide range of locations due to differences in counting methods, which is a common limitation to visitation records to natural spaces, generally. Capturing visitation to natural areas is very challenging, and existing approaches all have their own limitations for capturing daily visitation [40]. The observational data that overlaps with the cell data consists of: 1) onsite counts of small access points to an estuary, 2) a town's visitation estimates for their managed beaches, and 3) entrance fees collected by a town to a major beach.

Visitation observation methods also vary because the context for taking them differ. Although the objective may be the same, observing visitation at a major beach requires different methods than those used for a set of small access points around an estuary [41]. By comparing and calibrating our data to a combined set representing the variety of recreational visitation count methods that exist in the real world, we show the ability of cell data to both replicate the types of data that are traditionally used and to bridge the various observational visitation records common around natural resources. Each source of observational data is briefly described below.

- **Small:** We quantified use of the Three Bays estuary system on Cape Cod, Massachusetts through observational sampling for eleven public access points within the estuary. The counts were taken as a combination of periodic people and car counts and sunrise-to-sunset counts of visitors and cars. The public access points include beaches, docks, boat ramps, and landings. Observational data from this study include visitation estimates for 11 public access points for seven days from June to August, 2017 [41]. (N = 72)

- **Medium:** The diverse set of beaches for Barnstable, Massachusetts, on Cape Cod includes saltwater and freshwater beaches accessible to either the public or to town residents only. The dataset provided by the Town of Barnstable's Recreation Division includes daily visitation estimates from lifeguard counts for seven of their beaches from Memorial Day (May 29) to Labor Day (September 4), 2017. (N = 234)

- **Large:** Narragansett Town Beach in Rhode Island is a popular destination for tourists and residents. Access to this beach is divided into resident only and public entrances. The public entrance requires an entrance fee for each person providing an accurate dataset of daily visitation to the beach. However, these entrance fees are only collected for the public part of the beach. By missing those visitors with resident permits, the data provided by the town of Narragansett underestimates visitation to the whole beach. Therefore, based on a .85/1 ratio between public and resident use obtained through parking lot counts conducted in the

public and resident parking lots, the data were adjusted to represent daily use for the whole beach (see S1 File for details). Daily visitation numbers are provided from Memorial Day to Labor Day, 2017. (N = 86)

Fig 2 shows how the cell data-derived counts and observational counts compare. The cell data-derived estimates correspond well, but overestimate the observed visitor counts by about four times. This overestimation is likely due to several confounding assumptions. These assumptions start with choices in how the data provider processes the raw cellular device level information to associate records with geographies in certain time windows and to extrapolate the sample of cellular devices to population level estimates through their estimates of market penetration (see S1 File).

The cell data product did not provide counts for the block of time that corresponded to our visitation counts (9AM-4PM), but rather by individual hours. Therefore, we had to translate these hourly counts to our time window by making assumptions on the length of stay of visitors, since the same device would be counted multiple times if it were to stay at the site for multiple hours. Following the data provider's advice to match the cell data-derived information to visitation observations, we used an assumption of a three-hour average stay to match the time-

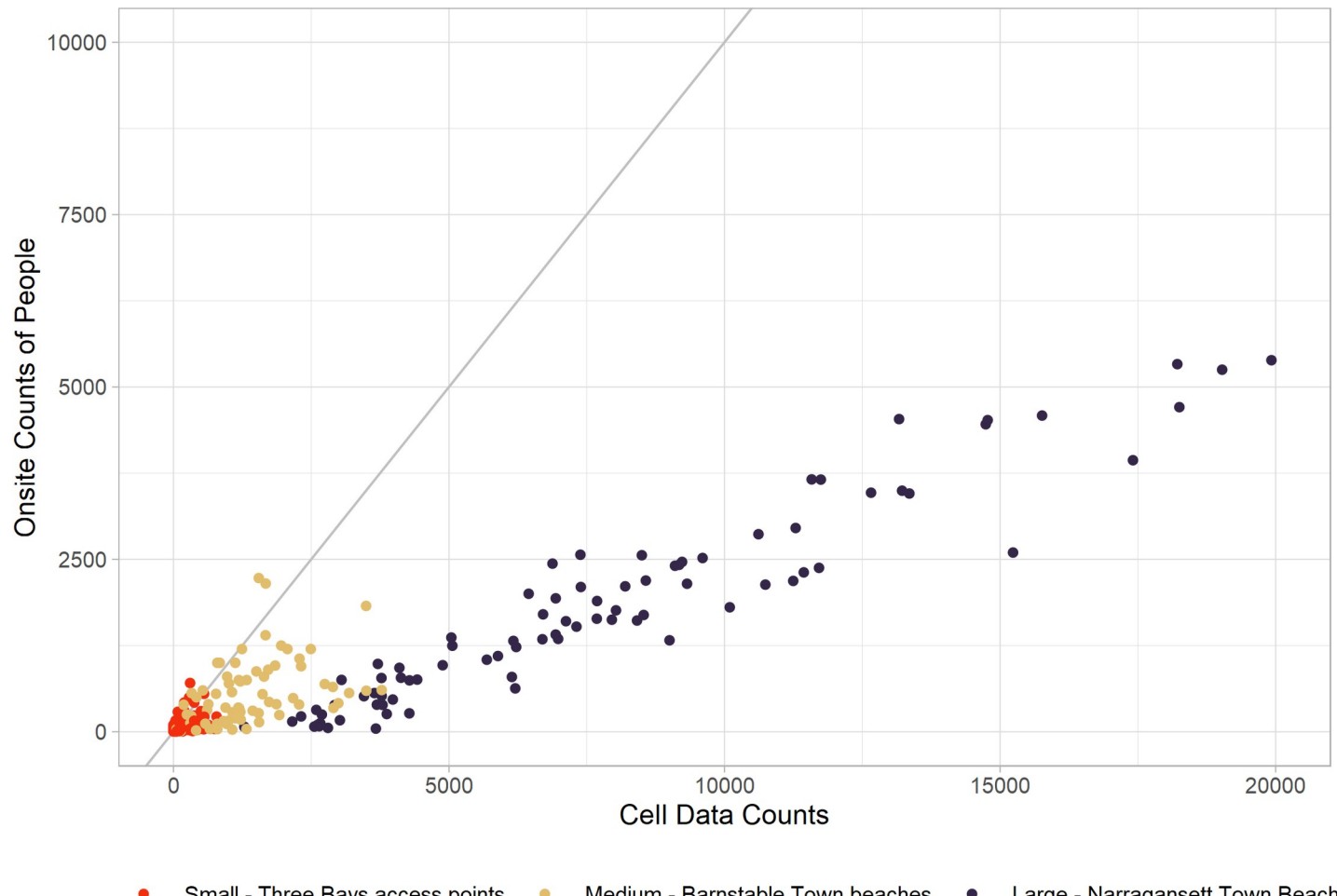

**Fig 2. Observational counts compared against cell data.** Observations of visitation (9AM-4PM) plotted against uncalibrated visitation estimated from the cell data for the same hours.

window of our observations. We could have picked data on one hour in the window to be representative of the whole three hours, the end hour for instance, but this would discard information in the other hours. Instead, we calculated a moving average (three-hour window) of visitation for each hourly visitation estimate from the cell data for each site. We then summed the moving average of only the central hour of three-hour blocks from 8AM-4PM (9AM, 12PM, 3PM) (see S1 File for more details). This reduces the cell data counts due to multiple sightings, since we summed only one of the three hours in each window, but maintains the information in the hourly distribution of use through the day.

An assumption of a shorter length of stay would have increased the cell data counts and vice-versa. For example, if we assumed a two-hour average length of stay, we would have used two-hour instead of three-hour windows in the daily sums, increasing the daily total. While three hours may be a long average length of stay for recreational visits to all the water access sites, the data reflecting this assumption were inputs to the calibration models. We sought to correct any bias and inaccuracies introduced by this assumption by using the calibration models fit to on-the-ground counts below. Similarly, there is a difference in the relationship across the three sizes of access points that can also be seen in Fig 2. The differences by group are likely due to differences in the observational counting methods and possibly how well cell data performs based on the size of the area. We control for both possible effects in the statistical models used to calibrate the data.

## Models and prediction

Our objective was to develop a model that predicts visitation to a range of water recreation areas using the cell data and other explanatory variables that are easily compiled across many places. These covariates include weather (temperature and precipitation), the month, day of the week, and size of the water access. The model controls for the different counting methods in the observational data. We estimated a varied set of candidate regression models including several functional forms where we defined linear and log-linear relationships between the visitor counts and the cell data and other regressors in R [42]. Since we did not have any preconceived notion of the functional forms of the relationships between the covariates and the dependent variable, we also estimated a more general random forest model. A random forest model is a type of non-parametric model commonly used in the data science and machine learning fields. They have been shown to reproduce many functional forms and have superior predictive performance over standard multivariate regression models for many applications [43–44].

The candidate model specifications were as follows:

- Linear

$$Y_{it} = \propto + \beta_1 C_{it} + \boldsymbol{\beta_2 D_t} + \boldsymbol{\beta_3 W_t} + \boldsymbol{\beta_4 S_i} + \beta_5 A_i + e_{it} \tag{1}$$

- Log-Linear

$$\log(Y_{it}) = \propto + \beta_1 C_{it} + \boldsymbol{\beta_2 D_t} + \boldsymbol{\beta_3 W_t} + \boldsymbol{\beta_4 S_i} + \beta_5 A_i + e_{it} \tag{2}$$

- Random Forest

$$Y_{it} = f(C_{it}, \boldsymbol{D_t}, \boldsymbol{W_t}, \boldsymbol{S_i}, \boldsymbol{A_i}) \tag{3}$$

Where,

$Y_{it}$ - Observed visits to site, $i$, on day, $t$.

$C_{it}$ - cell data-derived estimate of visitation to site $i$ on day $t$.

$D_t$ - Matrix of dummy variables for the month, day of the week, weekend, holiday.

$W_t$ - Matrix of weather variables (precipitation, temperature, windspeed) for day $t$ from Barnstable Municipal Airport weather station.

$S_i$ - Dummy variables for the source of observed visitation data (Narragansett Beach, Barnstable Town, Three Bays).

$A_i$ - Area of site $i$.

$\propto$ - intercept.

$e_{it}$ - error term.

We compared the candidate models based on their predictive performance. To avoid selecting an overfit model and therefore being overconfident in its out-of-sample model performance, we conducted a cross validation by splitting the data into training (in-sample) and test (out-of-sample) sets. We fit the candidate models to the training sets of data, predicted the test sets, and calculated fit statistics on the out-of-sample observations. We did this for 10 random splits of the data using a k-fold cross validation and present the average model performances across the 10 test sets in Table 1 [45–46]. Using this cross-validation technique is a statistical check and a data science best practice. Given the predictive purposes of these classes of models, we suggest that, in the future, a similar cross-validation process should be performed when using proxy-types of data for predicting visitation. Additional regression and random forest outputs, goodness-of-fit metrics, and details can be found in the S1 File as well as in the code package (https://github.com/USEPA/Recreation_Benefits.git).

## Ethics statement

This work was reviewed and deemed exempt by the Institutional Review Board (Study #17–3334) from the University of North Carolina at Chapel Hill.

## Results

Using the cell data product from the data provider resulted in about a four-times overestimation of the type of recreational visitation we were looking to estimate when compared against observations. Despite the scale difference, we found the information contained in the cell data to be valuable to predict visitation across a diverse set of sites after calibration. From our models, there are a few ways to show that it is the information in the cell data that is providing most of the explanatory power as compared to the covariates (weather, area, source of the observational counts). Table 2 shows the regression results using just cell data (columns 1–2), then with additional covariates (columns 3–4). Just using cell data produces a decent model in linear form. The additional value of the covariates can be seen in the improved stats between column 1 and 3 and 2 and 4. In the random forest model, the cell data was by far the most useful in modeling visitation as seen by metrics of variable importance (see S1 File)

Of the candidate models, we chose the random forest as the preferred model, given its predictive performance indicated by the lowest RMSE, MAE, and low bias (ME) during cross validation (see Table 3). To create the final comprehensive visitation dataset, we used the preferred random forest model to predict daily visitation to all 577 water-access areas in our sample for the four summer months (June-September) of 2017. Fig 3 plots predicted visitation using the random forest model against observational counts, showing a tight overall in-sample fit. Our calibrated model produced daily visitation estimates with an out-of-sample mean absolute error of 155 people (Table 3) based on the cross-validation.

**Table 2. Regression results.**

|  | Visits | Log(Visits) | Visits | Log(Visits) |
|---|---|---|---|---|
|  | (1) | (2) | (3) | (4) |
| Cell data | 0.245*** | 0.0003*** | 0.296*** | 0.0002*** |
|  | (0.005) | (0.0000) | (0.007) | (0.0000) |
| Area (m$^2$) |  |  | 0.00003 | 0.000006*** |
|  |  |  | (0.0002) | (0.0000006) |
| Narragansett |  |  | -646.796*** | 0.182 |
|  |  |  | (72.136) | (0.225) |
| Town of Barnstable |  |  | -60.612 | -0.409*** |
|  |  |  | (42.119) | (0.131) |
| Temperature (˚F) |  |  | 10.398*** | 0.066*** |
|  |  |  | (2.567) | (0.008) |
| Precipitation (inches) |  |  | -26.180 | -0.447*** |
|  |  |  | (47.250) | (0.147) |
| Constant | 33.128* | 4.320*** | -334.114 | 0.539 |
|  | (19.496) | (0.064) | (206.779) | (0.644) |
| Observations | 392 | 392 | 392 | 392 |
| R$^2$ | 0.86 | NA | .89 | NA |
| ME | .27 | -430.96 | .43 | -74.35 |
| RMSE | 318.08 | 3161.23 | 272.47 | 1030.21 |
| MAE | 186.69 | 791.33 | 174.68 | 345.93 |

Dummy variables are included for month and day of the week in columns 3 and 4. Columns 2 and 4 are in log-linear form. See code and S1 File for additional details and candidate models. Goodness of fit statistics are from out-of-sample sets from a 10-fold cross validation. Log-linear model predictions were converted to people terms for goodness of fit statistics. ME = mean error, RMSE = root mean squared error, MAE = mean absolute error.

* p<0.10.

** p<0.05p.

***p<0.01.

This model produced comprehensive visitation estimates across the region, but also at each individual site across days. The result of this work, a calibrated dataset of visitation including the code and data for producing the results in this paper can be found at https://github.com/USEPA/Recreation_Benefits.git. As examples of the usefulness of the broad geographical scope of the model and resulting database, Fig 4 shows daily visitation estimates to all public water-access points on Cape Cod for the summer months of 2017. Along with these types of landscape-scale results, the data and model provide focused site-specific information. Fig 5 shows daily visitation to a single beach across the season, information available for all sites and days across the four summer months.

**Table 3. Performance statistics for candidate models.**

|  | ME | RMSE | MAE | R-Squared |
|---|---|---|---|---|
| Linear model | 0.43 | 272.47 | 174.68 | .89 |
| Log-linear model | -74.35 | 1030.21 | 345.93 | NA |
| Random forest | -3.78 | 262.48 | 154.84 | .91 |

ME = mean error, RMSE = root mean squared error, MAE = mean absolute error. See S1 File for complete set of model output and performance statistics.

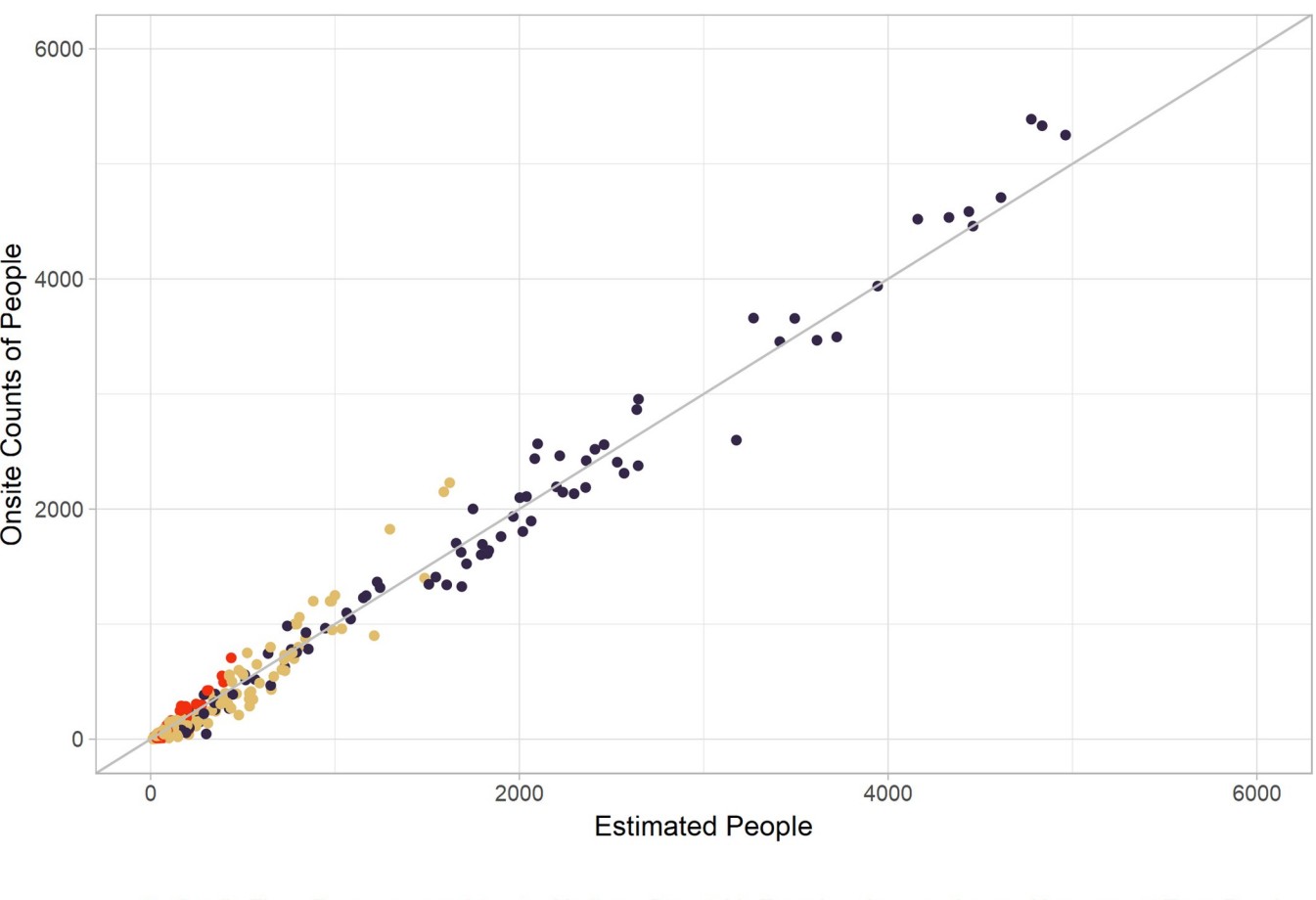

**Fig 3. Cell data modeled visitation compared against observations.** Predicted daily (9AM-4PM) visits from the cell data model compared to observed visitation at three sizes of water recreation areas in New England, USA.

Since we combined three different sources of visitation data to fit the model, we also ran candidate models on each visitation observation data source separately and the relationships between the cell data and each visitation dataset remained similar (see S1 File). The in-sample fit of those models varied, with the smaller access point dataset, Three Bays, the least well fit ($R^2 = .36$), to better fits with the larger access point of Narragansett Beach ($R^2 = .96$). The number of observations vary with the sources, as do specifics of how those observations were collected, but we suspect the cell data may be better at predicting visitation to larger areas, with more daily visitation. There are more cellular devices in a sample of a day at the more popular places to estimate visitation from, likely reducing noise in the estimate.

We used the most accurate and unbiased of the candidate statistical models for prediction. However, there are several sources of potential inaccuracies and biases in estimating visitation in this way that are not incorporated in the metrics of model goodness-of-fit. The observational visitation counts contain their own uncertainties and potential biases based on their sampling design and counting methods. By calibrating and validating to those counts, we may be carrying over those issues to our estimates of visitation. In addition, the cellular data contains uncertainties resulting from the geospatial accuracy of the device locations, our geographic definition of the sites, and the methods of expansion from the device sample to population-level estimates. More applications using cell datasets are needed to understand

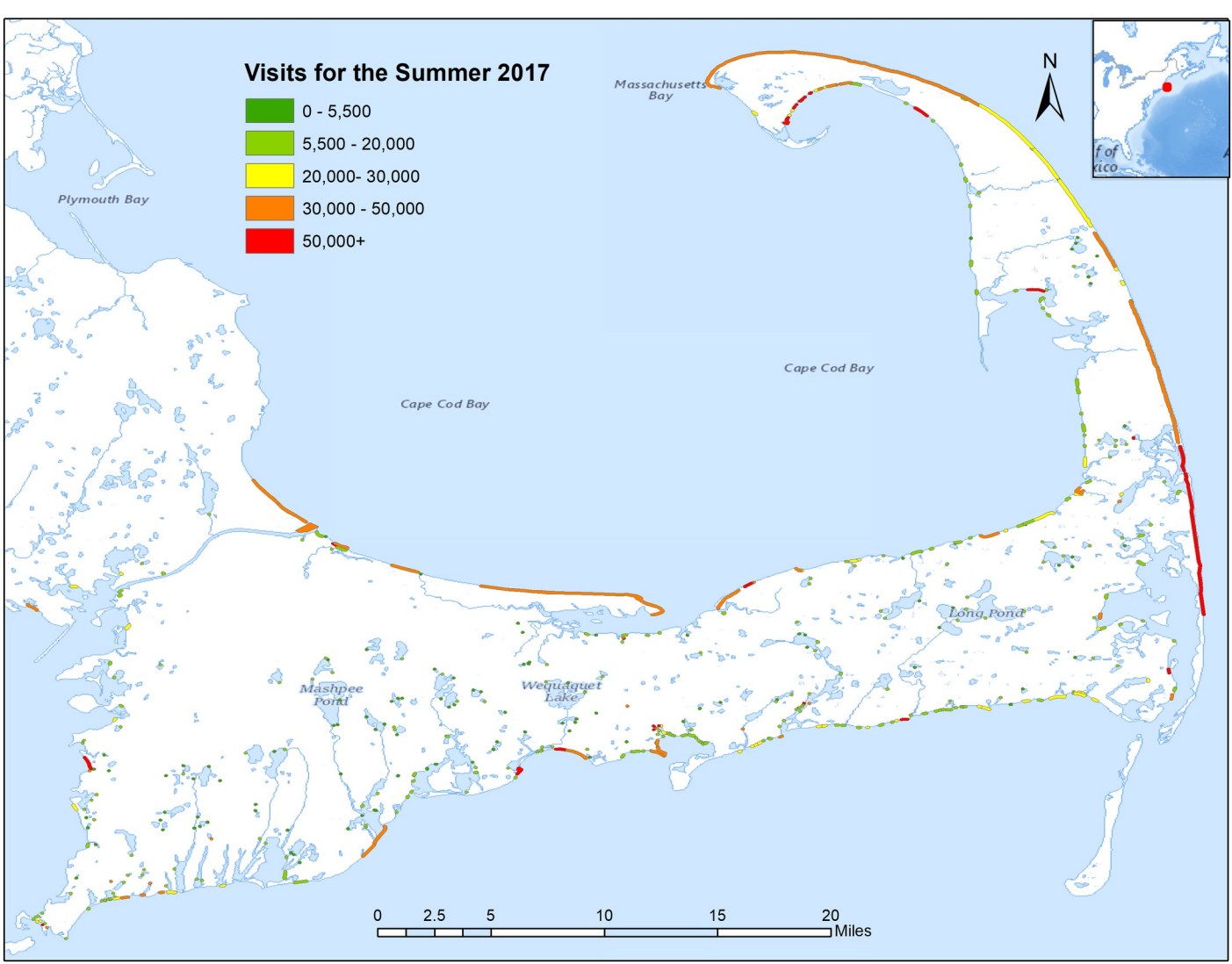

**Fig 4. Visitation for Cape Cod, MA, USA for the summer of 2017.** Total predicted visits (9AM-4PM) to water recreation areas for the summer of 2017 (June, July, August, September) for Cape Cod (Barnstable County, MA, USA), using the cell data model.

these limitations combined with additional and more consistent collections of visitation observations for calibration.

The models we fit may also be susceptible to spatial autocorrelation issues resulting from the cell dataset if there are variations in how the data represents visitation geographically. Spatial autocorrelation in models potentially inflates goodness-of-fit estimates, can bias parameters, and reduces predictive performance. We have controls in the models for each group of sites, which are geographically clustered to alleviate some of the potential issue. Similarly, collinearity in the covariates could potentially cause poor predictions, attributing predictive information to the wrong covariate, for example. We checked for this issue in a few ways by building up the model covariates sequentially starting with cell data alone and adding covariates. This led to little change in the relationships between observations and cell data counts (see S1 File for more model details and variations). We also consistently found good out-of-sample goodness-of-fit metrics in a cross validation, giving us more confidence that spatial

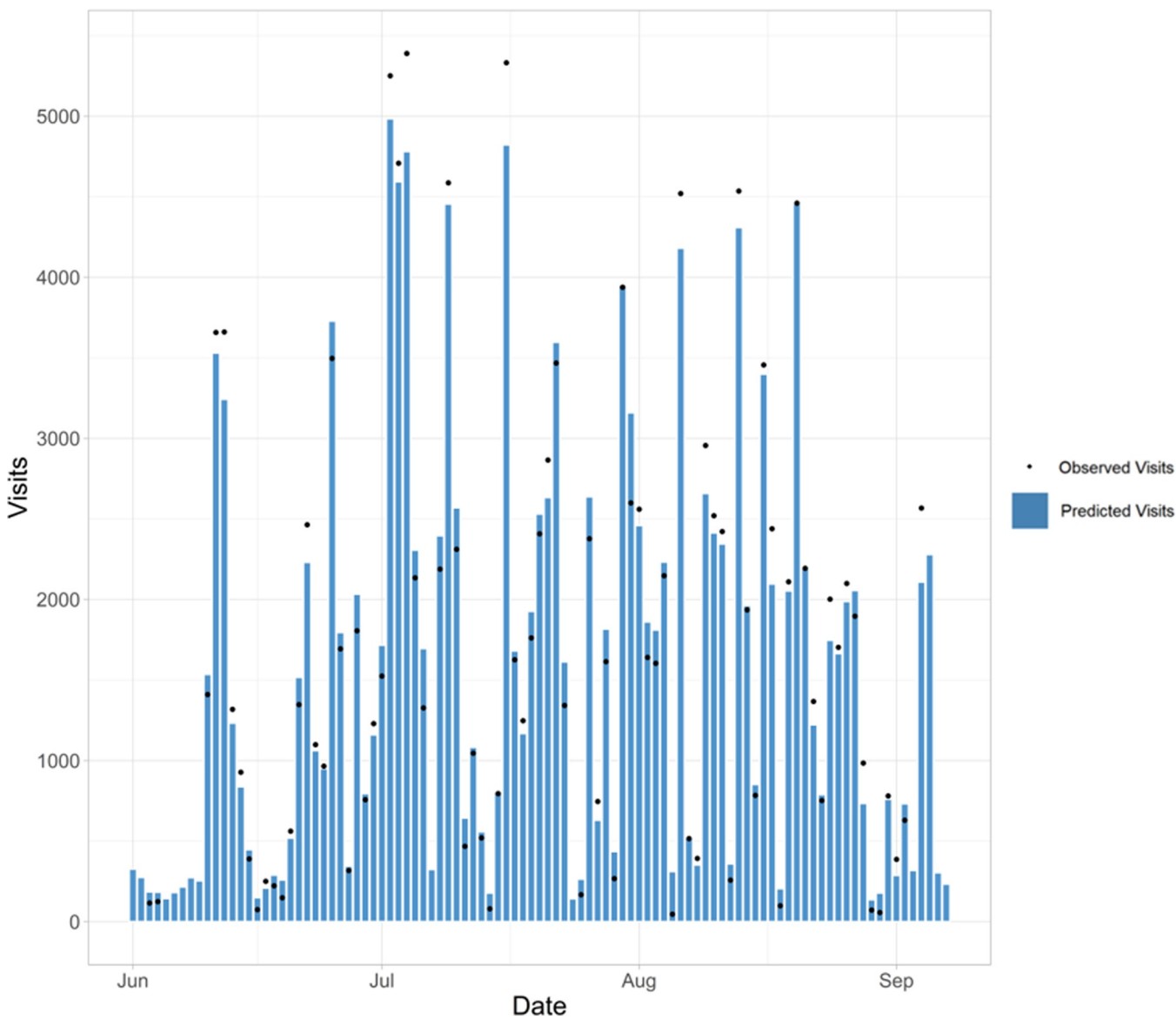

**Fig 5. Narragansett Beach, RI, USA, daily visitation.** Predicted daily (9AM-4PM) visits to Narragansett Town Beach (Narragansett, RI, USA) for the summer of 2017 using the cell data model. These daily predictions are compared to observed visits based on on-the-ground counts.

autocorrelation and collinearity were likely not an issue in the models' predictive performance.

## Discussion

We contend that the use of cell data provides a valuable method to quantify visitation across large numbers of areas and over time. When calibrated, the cell data provided an accurate and consistent way to estimate visitation to natural areas. It allowed us to produce a previously unavailable dataset of water visitation at a policy-relevant resolution, spatial extent, and consistency. The information provides visitation details for specific locations at a regional or sub-regional scale, the scale at which most decisions and policies regarding public natural areas are made. Understanding the scale and timing of visitor use of an area allows managers to

determine and provide appropriate facilities and safety precautions and researchers to predict impacts of environmental change, measure effects of natural disasters, and conduct research on social and economic value of public access to natural areas. To date, no other methods for quantifying use have the same capacity for providing location-specific data across broad geographic areas at such a high temporal resolution.

The data can inform landscape-level analyses of behavior to understand, at a broader scale, the impacts of changes in environmental quality across time as well as across access geography. The origin data can be used to better understand who visits an access point based on where they come from. This provides informative profiles regarding the characteristics of people and communities that are, or could be, affected by environmental degradation or improvements or other policy and management decisions. Taking advantage of variations in behavior and environmental quality across time, cell data's fine temporal resolution opens avenues for longitudinal studies. For instance, it provides the ability to quantify the number of visitors affected and economic impacts incurred from beach closures, algal blooms, oil spills, or other events that typically are not captured consistently with current methods or lack baseline data for measuring impacts.

Given our results, we suggest caution in using visitation estimates out-of-the-box from data providers without calibration. There is useful information in the datasets, but we found that calibration was necessary to confidently use the data for our purposes. In its delivered form, the data overestimated use of the type we were interested in quantifying, recreational visits to water-access areas. We hypothesized the need for this correction based on a few practical factors discussed below.

We were only able to define recreation visitation by limiting the area of the requested geographies (the GIS information for the sites we requested from the data providers) where recreation would likely be the primary purpose for visiting the area and during time-windows of interest. Inherently, the on-the-ground visitation observations were more restrictive in capturing visits for recreation and not capturing ancillary or non-recreational visits to the geographies, like walking by the site on a nearby road. It is also reasonable to assume some observational counts may be conservative and under-report recreation visitation due to sampling constraints.

Additionally, there is a cascade of statistical modeling assumptions that are made by the third-party providers to take raw device locational information to the anonymized and aggregated form delivered to the customers (see S1 File for the publicly available description of Airsage's process). The exact details of each private provider's data processing workflows are their intellectual property and protected as such. This opaqueness motivates the use of methods to judge performance critically and the construction of additional models based on common cell data products for popular environmental applications, such as the one described in this paper. Because the methods used by the data providers are constantly in development, estimating performance on common observational datasets and with common methods would provide more clarity and confidence (statistical and otherwise) for cell data's use in policy and management applications. In some cases, lack of cellular connectivity in some natural areas may also limit its usefulness, although GPS-based and application-derived locational methods have overcome some of this limitation by passing along information when the device is reconnected to a network. It is in a user's best interest to calibrate the product for their application, when possible, or consider the uncalibrated information as a relative metric. We demonstrated a simple method to do a calibration in this paper and provide the data and code for others to work from and improve as more users apply these types of data products.

While cell data-derived information is an exciting development for researchers and managers, counterintuitively, we found attempting to use it for a practical application only further

motivated the need to take more accurate, consistent and unbiased observations of visitation using traditional methods. Modeling methods are hindered by the lack of availability of training datasets and would be greatly improved by larger and more uniformly collected observations. This is especially true regarding machine-learning algorithms [47]. For example, with small and practical tweaks in the way visitation records are collected at water-access areas, such as collecting periodic counts of cars and people at specific times, visitation records could become more harmonized and useful [12,41]. From there, visitation proxies like cell data or social media-based models can provide a platform for spatial and temporal extrapolation across broad geographies, as we demonstrate here. The need for such models is not confined to water-access visitation, as it is relevant to many other similar policy contexts, for example, at national parks or urban green spaces. The differences in how well our models fit depending on the visitation data source, with larger more popular locations fit better than small, should be cautionary to approaches applying the cell data-derived visitation estimates to settings where there is no similar observational data for comparison.

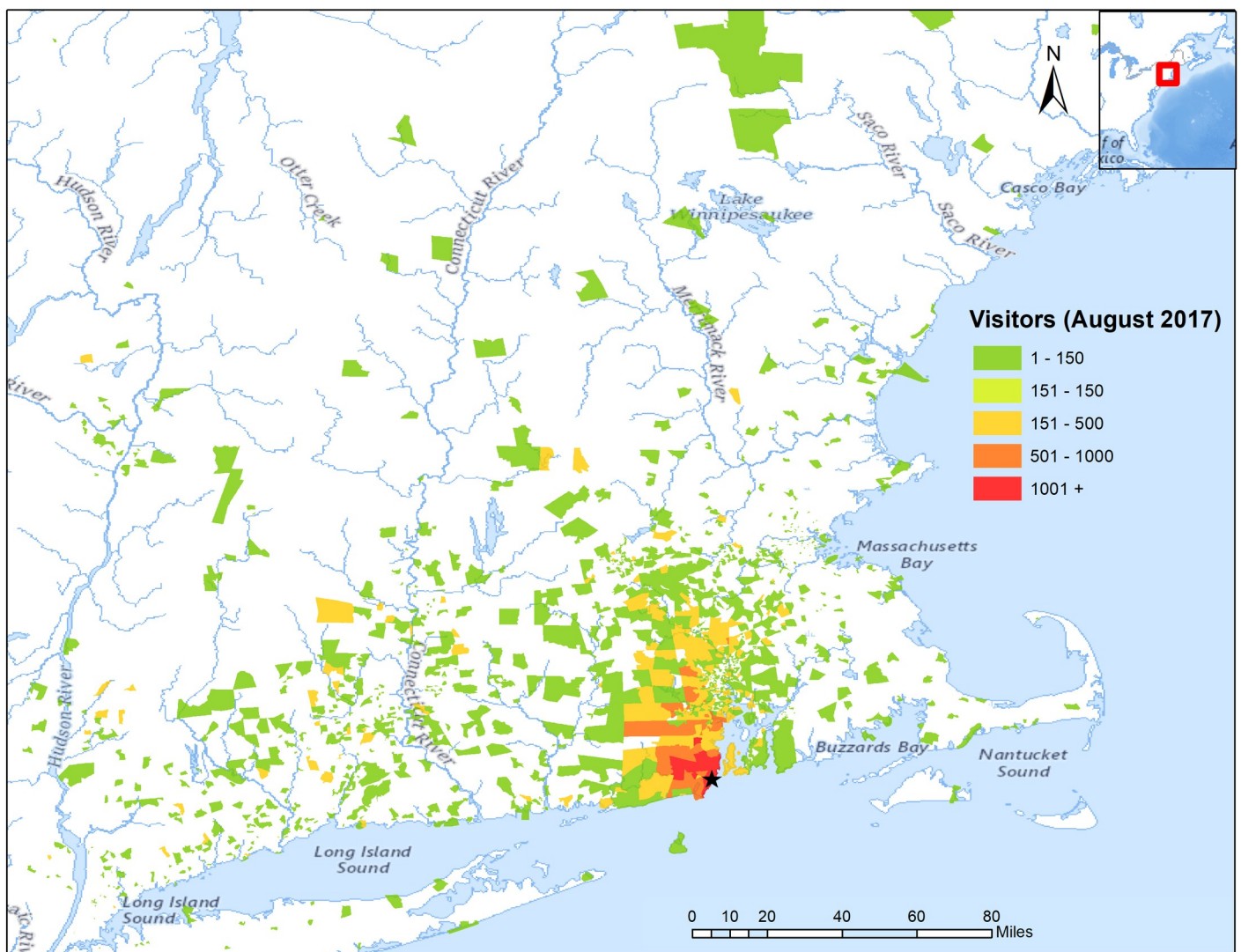

**Fig 6. Visitor origins for Narragansett Beach, RI, USA.** Count of visitors by census block group origins for visitors to Narragansett Town Beach, Narragansett RI, USA, (black star on the map) in August 2017. This monthly origin information exists for each of the 577 access points in our sample.

In this paper, we explored only a few dimensions of the information in the cell data-derived dataset, focusing on estimating visitation. The origin information provided about the visitors to the water-access sites—monthly visitation by site and census block groups in our case—provides the type of data needed for a myriad of travel-demand models for recreation [48]. Previous social media-based work [18] pointed out this potential opportunity, but travel-demand models typically require more spatially and temporally resolved information than what can be found currently in data derived from social media. Fig 6 displays origin-destination data for one site, Narragansett Beach, Rhode Island, for August 2017. Our dataset includes this information for all sites and months. More work is needed to calibrate and validate the origin-destination information provided by cell data products for environmental applications and more generally [49]. Validating the origin data and fitting travel-demand models is beyond the scope of this paper and is left for future work. We include the origin-destination data in the data package associated with this paper for any other users that wish to tackle this natural next step. As with the visitation estimates, the possibility of using new sources of data for origin-destination travel-demand models encourages implementation of more commonly worded and formatted general population and visitor intercept surveys to provide the necessary corroborating and calibrating datasets.

The visitation dataset resulting from the work in this paper is useful for federal, state and town managers and agencies in the region for any number of applications requiring visitor use information. For example, state managers may use this information to determine the allocation of beach monitoring resources, a town shellfish warden may use it to identify the most actively fished sites, or a tourism board could use it to understand the profile of visitors to popular natural attractions. Of broader interest in natural resource management and research, these types of models using cell data and a similar calibration process could be developed for other areas or for other purposes such as understanding the scale of ecosystem services, human-wildlife conflict, water consumption demand, or emergency response, for example.

## Supporting information

**S1 File.**
(DOCX)

## Acknowledgments

The views expressed in this article are those of the authors and do not necessarily represent the views or policies of the U.S. Environmental Protection Agency. This contribution is identified by tracking number ORD-028692 of the U.S. Environmental Protection Agency, Office of Research and Development, Center for Environmental Measurement and Modeling, Atlantic Coastal Environmental Sciences Division. We would like to thank Anne Neale, Bryan Milstead and Ryan Furey for their thoughtful reviews.

**Product disclaimer:** Mention of trade names or manufacturers does not imply U.S. Government endorsement of commercial products.

## Author Contributions

**Conceptualization:** Nathaniel H. Merrill, Sarina F. Atkinson, Kate K. Mulvaney, Marisa J. Mazzotta, Justin Bousquin.

**Data curation:** Nathaniel H. Merrill, Sarina F. Atkinson, Kate K. Mulvaney, Justin Bousquin.

**Formal analysis:** Nathaniel H. Merrill, Sarina F. Atkinson, Kate K. Mulvaney, Justin Bousquin.

**Investigation:** Nathaniel H. Merrill, Kate K. Mulvaney, Justin Bousquin.

**Methodology:** Nathaniel H. Merrill, Sarina F. Atkinson, Kate K. Mulvaney, Justin Bousquin.

**Project administration:** Nathaniel H. Merrill.

**Resources:** Kate K. Mulvaney.

**Software:** Nathaniel H. Merrill, Justin Bousquin.

**Supervision:** Nathaniel H. Merrill, Marisa J. Mazzotta.

**Validation:** Nathaniel H. Merrill, Sarina F. Atkinson, Kate K. Mulvaney, Justin Bousquin.

**Visualization:** Nathaniel H. Merrill, Kate K. Mulvaney, Justin Bousquin.

**Writing – original draft:** Nathaniel H. Merrill, Sarina F. Atkinson, Kate K. Mulvaney, Marisa J. Mazzotta, Justin Bousquin.

**Writing – review & editing:** Nathaniel H. Merrill, Sarina F. Atkinson, Kate K. Mulvaney, Marisa J. Mazzotta, Justin Bousquin.

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
