## [Decision Letter · Decision Letter 0]

14 Jan 2020

PONE-D-19-32712

Using Data Derived from Cellular Phone Locations to Estimate Visitation to Natural Areas: An Application to Water Recreation in New England, USA.

PLOS ONE

Dear Merrill,

Thank you for submitting your manuscript to PLOS ONE. After careful consideration, we feel that it has merit but didn't fully meet PLOS ONE’s publication criteria yet as it currently stands. Therefore, we invite you to submit a revised version of the manuscript that addresses the points raised during the review process.

We would appreciate receiving your revised manuscript by Feb 28 2020 11:59PM. To enhance the reproducibility of your results, we recommend that if applicable you deposit your laboratory protocols in protocols.io, where a protocol can be assigned its own identifier (DOI) such that it can be cited independently in the future. For instructions see: http://journals.plos.org/plosone/s/submission-guidelines#loc-laboratory-protocols

We look forward to receiving your revised manuscript.

Kind regards,

Professor Song Gao, Ph.D.

Academic Editor

PLOS ONE

Journal Requirements:

2. We note that Figures 3 and 5 in your submission contain map/satellite images which may be copyrighted.

a. You may seek permission from the original copyright holder of Figures 3 and 5 to publish the content specifically under the CC BY 4.0 license. 

3. Your ethics statement must appear in the Methods section of your manuscript. If your ethics statement is written in any section besides the Methods, please move it to the Methods section and delete it from any other section. Please also ensure that your ethics statement is included in your manuscript, as the ethics section of your online submission will not be published alongside your manuscript.

Reviewers' comments:

Reviewer's Responses to Questions

**Comments to the Author**

1. Is the manuscript technically sound, and do the data support the conclusions?

Reviewer #1: Yes

Reviewer #2: Partly

2. Has the statistical analysis been performed appropriately and rigorously? 

Reviewer #1: No

Reviewer #2: No

3. Have the authors made all data underlying the findings in their manuscript fully available?

Reviewer #1: Yes

Reviewer #2: No

4. Is the manuscript presented in an intelligible fashion and written in standard English?

Reviewer #1: Yes

Reviewer #2: Yes

5. Review Comments to the Author

Reviewer #1: This study presents a practical approach to estimate visitation to natural areas. While the overall methodology is sound, I do have some concerns in terms of how input data, which is used to fit the model, is derived:

1) The observation data are collected in very different means, which implies certain degree of inconsistency and systematic bias. It does make sense if the model is fit against a particular source of observational recording. However, I would question the validity if distinct observational sources are combined together. Have you tried to fit models against each individual data set and compare outcomes?

2) According to supplementary materials, to calibrate Narragansett beach observational counts, you calculated ratio (resident/public) in three 3-hour time windows and adopted the average (0.85) as the calibration ratio. The data suggests that the ratio is very different during different time windows, and the sum is actually very close 553 (public) vs. 528 (resident). Also, I am interested to know why three different dates (7/13, 7/24, and 7/26) were picked for data collection.

3) A spatial buffer of 100-meter was created and a reasonable explanation is provided. What role does locational accuracy play here? According to data description, location can be collected by GPS, Wi-Fi, or cell tower, and which location provider of each ping is unknown. For locations derived from cell tower, apparently the 100-meter threshold would lead to significant misclassification (i.e., inside or outside). You may want to justify the threshold selection by taking locational accuracy into account.

Some other questions/comments/suggestions:

1) The data description section can be improved. You mentioned "we obtain aggregated summaries of visitation by recreation site", what information does Airsage need (e.g., the name of the site, or the geographic extent of areas of interest) to generate aggregated summary and what do the "aggregated summaries" look like? I strongly recommend to include a table of sample data in this section.

2) How would applying 3-hour moving average to the raw hourly count address double counting (or multiple counting)? This is not very clear to me. Can you come up with better explanation in text?

Reviewer #2: This manuscript focuses on estimating the number of visitors to natural areas based on cellular data. The authors conducted multi-scale case studies based on cellular data and various field data (e.g., observed visitor data).

There have been abundant studies using cell phone data to study human mobility, but many previous studies were conducted from a perspective of transportation/urban geography. This study has a different angle and investigated the visitation pattern to natural areas based on cell phone data, which can potentially provide useful input to studies on natural resources in physical geography. To this end, it has the potential of becoming something useful in the field. However, several problems should be addressed before the authors move forward.

The structure of the paper can be improved. I would suggest moving the literature review in “data description” to the “background” section and keep the data description more focused. The literature review is generally inadequate. Please consider adding more details regarding the types of cell phone data (e.g., CDR, assisted GPS data, Erlang data) and how these datasets have been used to model human mobility. Another thing can be discussed is the connection and differences between using mobile phone data in physical geography and human/urban geography. The research question should also be explicitly stated in the introduction. Currently, it is unclear if the authors wanted to focus on analyzing the spatial distribution of visitations, the temporal pattern of visitations, or something else (although this information was later provided in the methodology section).

Related papers:

Calabrese, F., Ferrari, L. and Blondel, V. D. (2015). Urban sensing using mobile phone network data: a survey of research. ACM Computing Surveys 47(2), pp. 1–20.

Yuan, Y., & Raubal, M. (2016). Exploring Georeferenced Mobile Phone Datasets – A Survey and Reference Framework. Geography Compass, 10(6), 239-252. doi:10.1111/gec3.12269.

The authors should also discuss how their work is different from the following study:

MONZ, Christopher et al. Using Mobile Device Data to Estimate Visitation in Parks and Protected Areas: An Example from the Nature Reserve of Orange County, California. Journal of Park and Recreation Administration, [S.l.], v. 37, n. 4, oct. 2019. ISSN 2160-6862.

P4 l114, the definition of a “visit” is unclear. What if a user showed up at two nearby locations? For example, if a user sit on a bench for 20 minutes and then used a public restroom 100 meters away. Is this considered two visits? Doesn’t it make more sense to cluster close-by points from the same device?

The mobile phone dataset should also be better explained. It may help to provide a few sample records.

L198 how did you decide on the 100-meter buffer? Please clarify.

L248 Did you consider the collinearity between the explanatory variables in your model? Some of the explanatory variables may not be independent from each other. More importantly, it is highly likely that the observed values are spatially auto-correlated, which can inflate the R2 values and jeopardize the reliability of the models.

Would the results be different if you consider different time periods during the day/week/month, etc.?

Overall, the results are interesting but I feel that the models should be designed more thoroughly and carefully.

6. PLOS authors have the option to publish the peer review history of their article (what does this mean?). If published, this will include your full peer review and any attached files.

Reviewer #1: No

Reviewer #2: No

---

## [Author Response · Author response to Decision Letter 0]

20 Feb 2020

Reviewer 1:

1) The observation data are collected in very different means, which implies certain degree of inconsistency and systematic bias. It does make sense if the model is fit against a particular source of observational recording. However, I would question the validity if distinct observational sources are combined together. Have you tried to fit models against each individual data set and compare outcomes?

The issue of a lack of visitation data collection consistently and across many spaces and times is precisely the problem that we would hope the use of the cell data-derived estimates would overcome. However, as you point out, we are limited by not having perfect calibration/validation datasets. We point this out a few different ways in the paper [starting on line 71 and line 182, for example] and a paragraph in the discussion [starting on line 404]. Since we want to see how well the cell data works across many types and sizes of locations or visitation, we had to use observational records from many different sizes and types of locations. The observation methods vary because the settings for taking them differ. Even if the final objective of the sampling is the same, daily visitation, observing visitation at a big beach with car and people receipts calls for a different method than at a set of small access points around an estuary, for example. The way the description of the observational data was written implied we chose different count methods on purpose; it was also out of necessity to cover more types and number of places. We edited the description of the visitation datasets to make these points starting on line 182. 

We had to choose between using fewer observations from fewer places to fit our model, or more diverse observations from a larger set and more varied geographic settings. The latter is closer to the objective of the application of the cell data, so we chose that path. We sought to control for some of these differences in sources in the overall (all data source models) with a dummy variable for the source of data. 

With all that said, when fit separately, the observational data sources result in similar models. We now provide fits of models against each data source separately in the Supporting Information and discuss in the text. The observational count relationship to the cell data is consistent across the visitation data sources. They do vary with the coefficient related to the cell data ranging from .27-.32, but that is to be expected when cutting any dataset into parts and fitting separate models. The fit is best for Narragansett Beach followed by Barnstable’s town records then the Three Bays counts. This also happens to be in order of largest to smallest visitation locations. This leads us to hypothesize that there may be a varied accuracy by the quantity of visitation, possibly due to having a larger sample of devices in the datasets for larger beaches to represent daily visitation on any given day. We make note of these between observational dataset differences in the results and discussion:

Line 349:

“Since we combined three different sources of visitation data to fit the model, we also ran candidate models on each visitation observation data source separately and the relationships between the cell data and each visitation dataset remained similar (see Supporting Information). The in-sample fit of those models varied, with the smaller access point dataset, Three Bays, the least well fit (R2 = .36), to better fits with the larger access point of Narragansett Beach (R2 = .96). The number of observations vary with the sources, as do specifics of how those observations were collected, but we suspect the cell data may be better at predicting visitation to larger areas, with more daily visitation. There are likely more cellular devices in a sample of a day at the more popular places to estimate visitation from, reducing noise in the estimate.”

 And line 413:

“The differences in how well our models fit depending on the visitation data source, with larger more popular locations fit better than small, should be cautionary to approaches applying the cell data-derived visitation estimates to settings where there is no similar observational data for comparison.”

2) According to Supplementary Information, to calibrate Narragansett beach observational counts, you calculated ratio (resident/public) in three 3-hour time windows and adopted the average (0.85) as the calibration ratio. The data suggests that the ratio is very different during different time windows, and the sum is actually very close 553 (public) vs. 528 (resident). Also, I am interested to know why three different dates (7/13, 7/24, and 7/26) were picked for data collection.

We chose these time windows because from our work studying car/people methods on the Three Bays counting project (the small visitation dataset in this paper), the hours of 12-4 were most representative of the visitation for the day. The days were chosen out of convenience in the workweek over the summer. While Narragansett is unique in that it charges people (whether they park or not) to get on the beach, we found those records incomplete and chose this car method to capture the resident use. Each counting method comes with its own issues, which is another argument to include more than one observational dataset to compare results and calibrate more general models, such as the one we created.

3) A spatial buffer of 100-meter was created and a reasonable explanation is provided. What role does locational accuracy play here? According to data description, location can be collected by GPS, Wi-Fi, or cell tower, and which location provider of each ping is unknown. For locations derived from cell tower, apparently the 100-meter threshold would lead to significant misclassification (i.e., inside or outside). You may want to justify the threshold selection by taking locational accuracy into account.

This choice was one of balancing minimizing extraneous visitation to areas near, but not associated with, the access points of interest, but having areas big enough to represent the use there. Often, access point databases are in line or point form, so there has to be some buffer added to capture use around the area. We now include an example figure of these buffers (Fig 1). 

We tried 100-200-300-meter buffers and chose 100 in consultation with Airsage, where we believed we did not capture nearby attractions not wanted, but big enough for their process to work at the individual device level, which we as a buyer of the data do not see. Errors would go both ways: visit defined outside the area but actually in the area, or defined in the area but actually outside. We decided defining the area more narrowly as opposed to larger would limit the false attribution of visitation, which we figure is a worse direction for an error than being conservative and under-estimating visitation. 

The approach seemed to work well against observational counts, which is the only way we could assess this issue without access to individual level locational information. How big the buffer should be for what application is an interesting question that the data providers with access to the individual level data and locational processing methods could tackle. For us, the justification is showing the accuracy of our choices against the observational counts we are trying to calibrate to.

We re-worked this discussion now in the paper, line 136:

 “To obtain the cell data for the sample geographies of interest, we spatially buffered (added area) around the water-access sites which were designated as line or point features in the original spatial databases. In consultation with the data provider and after attempting a range of spatial buffers, a 100-meter buffer was chosen to balance specificity in capturing water recreation visits (i.e., not capturing ancillary points of interest in geographies, like restaurants or stores, for example) with the accuracy of the locational information. We sent the defined water recreation areas to the data provider as a set of geographic extents, or polygons (see Fig 1 for examples of area definitions), and they returned the aggregated and anonymized processed data in tabular form. We include a sample of the dataset below (Table 1) and include the entirety of this dataset available with the code package associated with this work at https://github.com/USEPA/Recreation_Benefits.git.

The locational accuracy of the device locations underpinning the data range depending on the source device and the smartphone application. The accuracy of reported locations from applications varies with ranges of 1-10 meters (GPS), 20-200 meters (Wi-Fi), and 100-2000 meters (cell tower-based) based on the method(s) each application uses to locate each device. We were not able to obtain an average locational accuracy for devices seen in our geographies in our specific dataset since the smartphone applications do not report the exact location methods to the data provider and we do not receive device-specific locational information. Given the potential range in location accuracy, visits attributed to a water recreation area could have actually been to a nearby attraction, or vice-versa. We chose a relatively small buffer around the recreation areas to be conservative in defining the area attributed to use of the site and to minimize any mis-located visits. Given this and other limitations, we relied on the calibration and validation to on-the-ground visitation counts to assess the usefulness and accuracy of the cell data for our application and the choice of spatial definitions and buffers around sites.”

Some other questions/comments/suggestions:

1) The data description section can be improved. You mentioned "we obtain aggregated summaries of visitation by recreation site", what information does Airsage need (e.g., the name of the site, or the geographic extent of areas of interest) to generate aggregated summary and what do the "aggregated summaries" look like? I strongly recommend to include a table of sample data in this section.

Airsage, and other providers, need geographic areas (polygons) defined that represent the points of interest, POIs. We created these polygons ourselves and sent them to Airsage. There are other products where a user might define a large extent and get a rasterized (heatmap) style summary, but these were not the ones we pursued. 

We rearranged and added to the data description to address these questions and other reviewers’ comments, moving the lit review of these cell data applications to the background section and focusing on the specifics of our data purchase in the data description section. We also brought parts of the methods up to the data description that were explaining parts of the process of what we provide to Airsage and what they sent back (data description) versus what we did with it (methods).

We added a sample of the data in a table to the data description section as suggested and the whole thing is available with the code package. We now include an example of the spatial definitions in Fig 1.

2) How would applying 3-hour moving average to the raw hourly count address double counting (or multiple counting)? This is not very clear to me. Can you come up with better explanation in text?

The key to understand this is that we applied the three-hour moving average, but only summed one hour of the three (the middle hour) to get to the day total. This reduces the sum by roughly 1/3, but maintains the information in all three hours instead of picking the end, middle or start of the 3 hour chunk to represent the 3. We re-worked and added to this description starting on Line 229:

“The cell data product did not provide counts for the block of time that corresponded to our visitation counts (9AM-4PM), but rather by individual hours. Therefore, we had to translate these hourly counts to our time window by making assumptions on the length of stay, since the same device would be counted multiple times if it were to stay at the site for multiple hours. Following the data provider’s advice to match the cell data-derived information to visitation observations, we used an assumption of a three-hour average stay to match the time-window of our observations. We could have picked data on one hour in the window to be representative of the whole three hours, the end hour for instance, but this would discard information in the other hours. Instead, we calculated a moving average (three-hour window) of visitation for each hourly visitation estimate from the cell data for each site. We then summed the moving average of only the central hour of three-hour blocks from 8AM-4PM (9AM, 12PM, 3PM) (see Supporting Information for more details). This reduces the cell data counts due to multiple sightings, since we summed only one of the three hours in each window, but maintains the information in the hourly distribution of use through the day.

An assumption of a shorter length of stay would have increased the cell data counts and vice-versa. For example, if we assumed a two-hour average length of stay, we would have used two-hour instead of three-hour windows in the daily sums, increasing the daily total. While three hours may be a long average length of stay for recreational visits to all the water access sites, the data reflecting this assumption were inputs to the calibration models. We sought to correct any bias and inaccuracies introduced by this assumption by using the calibration models fit to on-the-ground counts below. Similarly, there is a difference in the relationship across the three sizes of access points that can also be seen in Fig 2. The differences by group are likely due to differences in the observational counting methods and possibly how well cell data performs based on the size of the area. We control for both possible effects in the statistical models used to calibrate the data.”

Reviewer 2

The structure of the paper can be improved. I would suggest moving the literature review in “data description” to the “background” section and keep the data description more focused.

We rearranged these sections, moving the lit review of these cell data applications to the background section and focusing on the specifics of our data purchase in the data description section. We also brought parts of the methods up to the data description that were explaining parts of the process of what we provide to Airsage and what they sent back (data description) versus what we did with the data (methods).

The literature review is generally inadequate. Please consider adding more details regarding the types of cell phone data (e.g., CDR, assisted GPS data, Erlang data) and how these datasets have been used to model human mobility. Another thing can be discussed is the connection and differences between using mobile phone data in physical geography and human/urban geography.

We edited this section and added more references that explain the locational data sources to the background section. The technicalities of specific data inputs for the whole range of cellular locational data is beyond the scope of this paper and better suited to be explained in the references we provided. We now provide a concise overview of the various sources and explain where the dataset we purchased came from, GPS location from smartphone applications, in the data description section.

We do not think a discussion about the difference between physical and human/urban geography provides any clarity to the reader in understanding what we did or to understanding the types and goals of past applications of cell data better than the groupings of applications we point out in the literature review. We have added several recent relevant studies to the literature review. 

The research question should also be explicitly stated in the introduction. Currently, it is unclear if the authors wanted to focus on analyzing the spatial distribution of visitations, the temporal pattern of visitations, or something else (although this information was later provided in the methodology section).

 We edited the statement addressing this point in the introduction:

 Line 51:

“We investigated how cell data performs in providing the types of visitation information needed in policy applications, information on the temporal and spatial distribution of visits to natural areas.”

Related papers:

Calabrese, F., Ferrari, L. and Blondel, V. D. (2015). Urban sensing using mobile phone network data: a survey of research. ACM Computing Surveys 47(2), pp. 1–20.

We added this reference to the list of references for use in urban and transportation research in the background section. 

Yuan, Y., & Raubal, M. (2016). Exploring Georeferenced Mobile Phone Datasets – A Survey and Reference Framework. Geography Compass, 10(6), 239-252. doi:10.1111/gec3.12269.

This one seemed repetitive to other references we had, so we did not include it. 

The authors should also discuss how their work is different from the following study:

Monz, Christopher et al. Using Mobile Device Data to Estimate Visitation in Parks and Protected Areas: An Example from the Nature Reserve of Orange County, California. Journal of Park and Recreation Administration, [S.l.], v. 37, n. 4, oct. 2019. ISSN 2160-6862.

This work was published after we initially submitted our paper, and since then we have been in touch with the authors. Our work does differ in important ways, using records from more than one location and counts representing people visitation instead of cars. We now cover this and other relevant recent work in an extended discussion in the background section on park visitation estimates

Line 102:

“There are a few recent applications of cell data to understand behavior in and around natural areas [37-39]. Kubo et al. [37] used cell data to calculate the economic value of coastal tourism across Japan, but provided no ground truth to the visitation information provided by the cell data. For an island park in Korea [38], Kim et al. applied cell data to analyze tradeoffs between visitation and biodiversity and showed decent correlations between the cell data and monthly estimates of visitation to several specific locations on the island. A study of parks in California, USA, is the closest to the work presented in this paper [39]. The study used a similarly processed cell data product from a third-party vendor to estimate daily park visitation. They calibrated the cell data with just one set of data, vehicle counts on a major nearby road, finding a unit-value correction factor. They then validated their estimates against a single park’s gate traffic and parking information. They found good agreement with their corrected cell data model and daily vehicle counts. From there, they used park-specific vehicle-to-people ratios to extrapolate to the number of visitors to the other twenty-one parks of interest. Our study differs by incorporating multiple visitation records representing counts of people to a wider set of locations: eighteen different water recreation areas. We find similar potential for this data source to provide useful, policy-relevant visitor use information at daily and site-level scales for water recreation areas.”

P4 l114, the definition of a “visit” is unclear. What if a user showed up at two nearby locations? For example, if a user sit on a bench for 20 minutes and then used a public restroom 100 meters away. Is this considered two visits? Doesn’t it make more sense to cluster close-by points from the same device?

Processing the individual-level information is done by the data provider Airsage, which uses its own methods to define a visit based on the individual device location history. They define a visit as a device staying for more than 5 minutes in the area during an hour. They define device totals (only count a visit once if leaves and enters in the hour), as well as activity points (which deal with the same device re-entering as you describe). We used the device totals to compare to visitation as this is closer to what we would want to count, a unique visit. Airsage uses clustering to define the device’s location. See the Supporting Information for the publicly available information on Airsage’s process. We point out in the paper, line 390:

“Additionally, there is a cascade of statistical modeling assumptions that are made by the third-party providers to take raw device locational information to the anonymized and aggregated form delivered to the customers (see Supporting Information for the publicly available description of Airsage’s process). The exact details of each private provider’s data processing workflows are their intellectual property and protected as such. This opaqueness motivates the use of methods to judge performance critically and the construction of additional models for popular environmental applications for use with common data products, such as the one described in this paper…”

The mobile phone dataset should also be better explained. It may help to provide a few sample records.

We rearranged and added to the data description to address these questions and other reviewer’s comments, moving the lit review of these cell data applications to the background section and focusing on the specifics of our data purchase in the data description section. We also brought parts of the methods up to the data description that were explaining parts of the process of what we provide to Airsage and what they sent back (data description) versus what we did with it (methods).

We now provide a sample of the data we used in Table 1.

L198 how did you decide on the 100-meter buffer? Please clarify.

This choice was one of balancing minimizing extraneous visitation to areas near, but not associated with, the access points of interest, but having areas big enough to represent the use there. Often, access point databases are in line or point form, so there has to be some buffer added to capture use around the area. We now include an example figure of these buffers (Fig 1). 

We tried 100-200-300-meter buffers and chose 100 in consultation with Airsage, where we believed we did not capture nearby attractions not wanted, but big enough for their process to work at the individual device level, which we as a buyer of the data do not see. Errors would go both ways: visit defined outside the area but actually in the area, or defined in the area but actually outside. We decided defining the area more narrowly as opposed to larger would limit the false attribution of visitation, which we figure is a worse direction for an error than being conservative and under-estimating visitation. 

The approach seemed to work well against observational counts, which is the only way we could assess this issue without access to individual level locational information. How big the buffer should be for what application is an interesting question that the data providers with access to the individual level data and locational processing methods could tackle. For us, the justification is showing the accuracy of our choices against the observational counts we are trying to calibrate to.

We re-worked this discussion now in the paper, line 136:

 “To obtain the cell data for the sample geographies of interest, we spatially buffered (added area) around the water-access sites which were designated as line or point features in the original spatial databases. In consultation with the data provider and after attempting a range of spatial buffers, a 100-meter buffer was chosen to balance specificity in capturing water recreation visits (i.e., not capturing ancillary points of interest in geographies, like restaurants or stores, for example) with the accuracy of the locational information. We sent the defined water recreation areas to the data provider as a set of geographic extents, or polygons (see Fig 1 for examples of area definitions), and they returned the aggregated and anonymized processed data in tabular form. We include a sample of the dataset below (Table 1) and include the entirety of this dataset available with the code package associated with this work at https://github.com/USEPA/Recreation_Benefits.git.

The locational accuracy of the device locations underpinning the data range depending on the source device and the smartphone application. The accuracy of reported locations from applications varies with ranges of 1-10 meters (GPS), 20-200 meters (Wi-Fi), and 100-2000 meters (cell tower-based) based on the method(s) each application uses to locate each device. We were not able to obtain an average locational accuracy for devices seen in our geographies in our specific dataset since the smartphone applications do not report the exact location methods to the data provider and we do not receive device-specific locational information. Given the potential range in location accuracy, visits attributed to a water recreation area could have actually been to a nearby attraction, or vice-versa. We chose a relatively small buffer around the recreation areas to be conservative in defining the area attributed to use of the site and to minimize any mis-located visits. Given this and other limitations, we relied on the calibration and validation to on-the-ground visitation counts to assess the usefulness and accuracy of the cell data for our application and the choice of spatial definitions and buffers around sites.”

L248 Did you consider the collinearity between the explanatory variables in your model? Some of the explanatory variables may not be independent from each other. More importantly, it is highly likely that the observed values are spatially auto-correlated, which can inflate the R2 values and jeopardize the reliability of the models.

We do not think collinearity is an issue with our regression. We are interested in the relationship of the observational counts to the cell data, which is varying with temperature, weather, day of the week, potentially clouding the marginal effect of each variable separately. Collinearity could be an issue in inflating coefficient variances and false attribution of effect to one or the other collinear regressor and could hinder out-of-sample performance. We can check for collinearity issues on the cell data coefficient, and the models in general, by specifying the regression without other covariates and add groups of them sequentially. We did this in the Supporting Information model performance tables. We also show the variable importance output for the random forest model, showing removing cell data at each node resulted in the largest increase in variance of all the variables. This implies that it is the cell data with the information and not an issue of collinearity with combinations of the other covariates, like day of the week and weather. We reference these checks in the text starting on line 301:

“From our models, there are a few ways to show that it is the information in the cell data that is providing most of the explanatory power as compared to the covariates (weather, area, source of the observational counts). Table 2 shows the regression results using just cell data (columns 1-2), then with additional covariates (columns 3-4). Just using cell data produces a decent model in linear form. The additional value of the covariates can be seen in the improved stats between column 1 and 3 and 2 and 4. From the random forest model, the cell data was by far the most useful in modeling visitation as seen by metrics of variable importance (see Supporting Information).”

Spatial auto-correlation could potentially be an issue, but we have a few controls for it. First, we include a dummy variable for the source of the observational counts. This also acts as a dummy the geographic areas of the beaches as the counts come from geographically clustered areas. We also use a cross validation technique to check for overfitting issues in our model for any number of reasons and present out-of-sample goodness of fit estimates. If there were severe collinearity or spatial/temporal autocorrelation that affected the predictive performance of our model we would have seen poor out-of-sample performance, which we do not see.

Would the results be different if you consider different time periods during the day/week/month, etc.?

We looked at a specific time window of visitation within the days, since this was the same time window for which our observational counts were representative (9AM-4PM). We include all days of the week and summer months (June-September) in our analysis, again based on the observational datasets. How well the cell data performs in other time periods outside of our summer season observational dataset we calibrated to is unknown. However, most water access in New England occurs in the summer months, and these are the most policy-relevant months to study. 

The cell data processing from Airsage is consistent year round, so we can assume the relationships would be similar, but we have no data to test that currently. At this point, we are confident in showing how cell data and our models do during the summer season in the region at these types of water access points in our sample.

---

## [Decision Letter · Decision Letter 1]

24 Mar 2020

PONE-D-19-32712R1

Using Data Derived from Cellular Phone Locations to Estimate Visitation to Natural Areas: An Application to Water Recreation in New England, USA.

PLOS ONE

Dear Merrill,

Thankss for your efforts and submitting your revised manuscript to PLOS ONE.  The revision has improved significantly. Before acceptance for publication, both expert reviewers requested a short discussion about the data quality, model uncertainty, and limitation among other minor issues. Therefore, we invite you to submit another revised version of the manuscript that addresses the minor points raised during the review process.

We would appreciate receiving your revised manuscript by May 08 2020 11:59PM. To enhance the reproducibility of your results, we recommend that if applicable you deposit your laboratory protocols in protocols.io, where a protocol can be assigned its own identifier (DOI) such that it can be cited independently in the future. For instructions see: http://journals.plos.org/plosone/s/submission-guidelines#loc-laboratory-protocols

We look forward to receiving your revised manuscript.

Kind regards,

Song Gao, Ph.D.

Academic Editor

PLOS ONE

Reviewers' comments:

Reviewer's Responses to Questions

**Comments to the Author**

1. If the authors have adequately addressed your comments raised in a previous round of review and you feel that this manuscript is now acceptable for publication, you may indicate that here to bypass the “Comments to the Author” section, enter your conflict of interest statement in the “Confidential to Editor” section, and submit your "Accept" recommendation.

Reviewer #1: All comments have been addressed

Reviewer #2: All comments have been addressed

2. Is the manuscript technically sound, and do the data support the conclusions?

Reviewer #1: Yes

Reviewer #2: Yes

3. Has the statistical analysis been performed appropriately and rigorously? 

Reviewer #1: Yes

Reviewer #2: Yes

4. Have the authors made all data underlying the findings in their manuscript fully available?

Reviewer #1: Yes

Reviewer #2: No

5. Is the manuscript presented in an intelligible fashion and written in standard English?

Reviewer #1: Yes

Reviewer #2: Yes

6. Review Comments to the Author

Reviewer #1: I really appreciate the efforts to address my comments for the original submission. The quality of this manuscript is significantly improved. I particularly like how you acknowledge and discuss the limitation of the data/method and how people should use caution when using cell data to address real-world problems.

Some trivial suggestions:

1) Can you name different types of data in term of data collection method, instead of "small", "medium", and "large"?

2) The ethics statement breaks the transition between methods and results. It should be moved to the end of the manuscript.

Reviewer #2: The authors addressed most of my comments in the revision, and the manuscript has been greatly improved. I only have one minor comment- I suggest adding a thorough discussion on the data quality issues (e.g., precision, accuracy, biases) that occurred in the study, as well as the uncertainty caused by modeling fitting. You can also clarify the collinearity and spatial autocorrelation issues in the main text and point the readers to the supplementary materials for more details.

7. PLOS authors have the option to publish the peer review history of their article (what does this mean?). If published, this will include your full peer review and any attached files.

Reviewer #1: No

Reviewer #2: No

---

## [Author Response · Author response to Decision Letter 1]

30 Mar 2020

Response to Reviewers 

Reviewer 1:

Reviewer #1: I really appreciate the efforts to address my comments for the original submission. The quality of this manuscript is significantly improved. I particularly like how you acknowledge and discuss the limitation of the data/method and how people should use caution when using cell data to address real-world problems.

Some trivial suggestions:

1) Can you name different types of data in term of data collection method, instead of "small", "medium", and "large"?

We considered this, but the current way was the most clear and concise way to refer to the different groups of coastal sites. The scale of the site and visitation we believe to also be an important distinguishing feature, especially in future use of the data. The size also corresponds to the counting method for practical observational reasons which we discuss in depth now starting on line 193.

2) The ethics statement breaks the transition between methods and results. It should be moved to the end of the manuscript.

It is an odd place for it, but when submitting our revisions, we were instructed by the editor to put it there:

“Your ethics statement must appear in the Methods section of your manuscript. If your ethics statement is written in any section besides the Methods, please move it to the Methods section and delete it from any other section.”

We will ask if we can put it at the end of the manuscript when we submit this version, but it might be the journal’s format to have it in the methods.

Reviewer 2

The authors addressed most of my comments in the revision, and the manuscript has been greatly improved. I only have one minor comment- I suggest adding a thorough discussion on the data quality issues (e.g., precision, accuracy, biases) that occurred in the study, as well as the uncertainty caused by modeling fitting. You can also clarify the collinearity and spatial autocorrelation issues in the main text and point the readers to the supplementary materials for more details.

In addition to the discussion of the accuracy issues in the cell data, we added a longer discussion in the results section about the compounding sources of bias and inaccuracies starting on line 359:

“We used the most accurate and unbiased of the candidate statistical models for prediction. However, there are several sources of potential inaccuracies and biases in estimating visitation in this way that are not incorporated in the metrics of model goodness-of-fit. The observational visitation counts contain their own uncertainties and potential biases based on their sampling design and counting methods. By calibrating and validating to those counts, we may be carrying over those issues to our estimates of visitation. In addition, the cellular data contains uncertainties resulting from the geospatial accuracy of the device locations, our geographic definition of the sites, and the methods of expansion from the device sample to population-level estimates. More applications using cell datasets are needed to understand these limitations combined with additional and more consistent collections of visitation observations for calibration. 

The models we fit may also be susceptible to spatial autocorrelation issues resulting from the cell dataset if there are variations in how the data represents visitation geographically. Spatial autocorrelation in models potentially inflates goodness-of-fit estimates, can bias parameters, and reduces predictive performance. We have controls in the models for each group of sites, which are geographically clustered to alleviate some of the potential issue. Similarly, collinearity in the covariates could potentially cause poor predictions, attributing predictive information to the wrong covariate, for example. We checked for this issue in a few ways by building up the model covariates sequentially starting with cell data alone and adding covariates. This led to little change in the relationships between observations and cell data counts (see Supporting Information for more model details and variations). We also consistently found good out-of-sample goodness-of-fit metrics in a cross validation, giving us more confidence that spatial autocorrelation and collinearity were likely not an issue in the models’ predictive performance.”

We also have a discussion point, pointing out the need for more and more consistent, unbiased collection of observational visitation data in traditional means. The lack of which holds back the new methods, like using cell datasets. Starting on line 424:

“While cell data-derived information is an exciting development for researchers and managers, counterintuitively, we found attempting to use it for a practical application only further motivated the need to take more accurate, consistent and unbiased observations of visitation using traditional methods. Modeling methods are hindered by the lack of availability of training datasets and would be greatly improved by larger and more uniformly collected observations. This is especially true regarding machine-learning algorithms [47]. For example, with small and practical tweaks in the way visitation records are collected at water-access areas, such as collecting periodic counts of cars and people at specific times, visitation records could become more harmonized and useful [12,41]. From there, visitation proxies like cell data or social media-based models can provide a platform for spatial and temporal extrapolation across broad geographies, as we demonstrate here. The need for such models is not confined to water-access visitation, as it is relevant to many other similar policy contexts, for example, at national parks or urban green spaces. The differences in how well our models fit depending on the visitation data source, with larger more popular locations fit better than small, should be cautionary to approaches applying the cell data-derived visitation estimates to settings where there is no similar observational data for comparison.”

---

## [Editor Report · Decision Letter 2]

3 Apr 2020

Using Data Derived from Cellular Phone Locations to Estimate Visitation to Natural Areas: An Application to Water Recreation in New England, USA.

PONE-D-19-32712R2

Dear Dr. Merrill,

We are pleased to inform you that your manuscript has been judged scientifically suitable for publication and will be formally accepted for publication once it complies with all outstanding technical requirements.

With kind regards,

Song Gao, Ph.D.

Academic Editor

PLOS ONE
---

## [Editor Report · Acceptance letter]

7 Apr 2020

PONE-D-19-32712R2 

Using Data Derived from Cellular Phone Locations to Estimate Visitation to Natural Areas: An Application to Water Recreation in New England, USA. 

Dear Dr. Merrill:

I am pleased to inform you that your manuscript has been deemed suitable for publication in PLOS ONE. Congratulations! Your manuscript is now with our production department. 

With kind regards,

on behalf of

Dr. Song Gao 

Academic Editor

PLOS ONE